# The group-housed pigs attacking and daily behaviors detection and tracking based on improved YOLOv5s and DeepSORT

**Tianyu Cheng[1], Fujie Sun[2], Liang Mao[3]*, Haoxuan Ou[2], Shuqin Tu[2]*, Fang Yuan[4], Hairan Yang[2]**

**1** Jiangxi Vocational College of Mechanical & Electrical Technology, Nanchang, China, **2** College of Mathematics and Informatics, South China Agricultural University, Guangzhou, China, **3** Undergraduate School of Artificial Intelligence, Shenzhen Polytechnic University, Shen Zhen, China, **4** College of Computer Science and Electronic Engineering, Hunan University, Changsha, China

* tsq5_6@scau.edu.cn (ST); maoliangscau@szpu.edu.cn (LM)











## Abstract

Automatic detection and tracking of pig behaviors through video surveillance remain challenges due to farm demanding conditions, e.g., illumination conditions and occlusion of one pig from another. The main goal of this study is to develop a deep learning method based on the improved YOLOv5s and DeepSORT to detect and track the behaviors of pigs, which has the advantages of stability and high accuracy. Firstly, YOLOv5s with the attention mechanism is used for pig detection and behavior recognition. To deal with the missed detection and false detection due to occlusion and overlapping between pigs and pigs, the improved YOLOv5s adopts the Shape-IoU to optimize the bounding box regression loss function, which improves the robustness of the model. Then, the improved DeepSORT model is proposed to track each pig behaviors including eat, stand, lie and attack four behavior types. Finally, we conduct a comparison test under different lighting and density conditions for pig detection and behavior tracking on special dataset. Experimental results show that the mAP@0.5% of improved YOLOv5s algorithm increases from 92.7% to 99.3%, which means 6.6% accuracy improvement compared with the YOLOv5s model. In terms of tracking, the values of MOTA and MOTP in all test videos are 94.5% and 94.9% respectively. These experiments demonstrate that the improved YOLOv5s and DeepSORT achieves high accuracy for both pig detection and behavior tracking. The proposed approach provides scalable technical support for contactless automatic pig monitoring.

## 1 Introduction

The accurate tracking of feeding and associated behaviors is an important challenge for the early detection of health and welfare challenges in livestock farming [1]. The

**Data availability statement:** The data has been saved in a public repository. URL:https://figshare.com/articles/dataset/paper_tu_data/29083367?file=54585815.

**Funding:** The author who has received the fundings is LM. The work was supported by Shenzhen Polytechnic University Smart Agriculture Innovation Application R&D Center (Grant NO.602431001PQ), Huizhou Municipal Key Areas Research and Development Project (Grant No.2024BQ010007), National Natural Science Foundation of China (Grant No.62272320). Shenzhen Science and Technology Innovation Commission Foundation (Grant No.20220812222043002) and Shenzhen Polytechnic University Research Fund (Grant No.6025310045K).

**Competing interests:** The authors have declared that no competing interests exist.

pig's behavior analysis is an important impact factor on the farm efficiency and the health [2]. The clinical or sub-clinical signs of most pig diseases are often associated with abnormal pig behavior before the diseases are found. Therefore, the monitoring and analysis of pig activity, diet, attack, and other behaviors can help to quickly understand the health condition of pigs. Regular monitoring of pigs' physical activity is also essential to identify short and long-term pig stresses [3].

Recent studies have used automatic surveillance techniques to observe pig behavior for early detection of potential health or welfare problems [4–6]. Among these techniques, YOLO detectors [7–9] and Multi-Object Tracking (MOT) [10–12] are the suitable approaches for detecting, classifying and tracking behaviors recognition due to theirs low cost and the implementation simplicity. Most of the existing MOT methods such as SORT [13] and DeepSORT [14] attempt to address the problem using the detector and tracker two separate models. The detector in these approaches is how to accurately detect object under scene of occlusion and variable light. For example, the CNN method was developed to classify different types of social behaviors among preweaning piglets: snout-snout and snout-body social nosing, and snout-snout and snout-body aggressive/playing behavior [15]. The CNN and LSTM were used to recognize the behavior of pigs, and detection accuracy of pig behavior reached 97.2% [16]. In reference [17], the YOLOv8 is employed for the real-time detection and behavior classification of pigs under variable light and occlusion scenes. All these approaches have achieved good results, but they do not consider the attention mechanism. Attention not only tells where to focus; it also improves the representation of interests. In this study, we fused the YOLOv5s and Convolutional Block Attention Module (CBAM) [18] to improve the detection performance by focusing on important features and suppressing unnecessary ones.

Nowadays, more and more research using MOT technology has been applied for the detection, counting and tracking of pigs. For example, a novel two-stage method [19] was proposed for automating the estimation of social cohesion in preweaning piglets while simultaneously detecting outliers. The proposed pose estimation model achieves a mean average precision (mAP) of 86.1% and 95.3% for object detection and pose estimation. In reference [20], a tracking method was developed for weaner pigs housed in partly covered pens, with the particular aim to re-identify individuals when they reappear in the field of view. The ORP-Byte MOT method [21] based on a rotated bounding box detector was proposed and achieved an MOT accuracy (MOTA) of 99.8%, an IDSW of 16, an identity F1-score (IDF1) of 91.6%. Based on graph convolutional networks, a robust computer vision algorithm [22] for long-term animal tracking was proposed for handling errors from the detector in challenging environments. Our previous work [10] proposed a YOLOv5-Byte MOT method for pig behavior analysis, which accurately monitored individual pig behaviors and provides statistical analysis. These methods focused on the daily condition of pigs, and little research was conducted on both pig attacking and daily behaviors tracking.

In this work, we have developed an improved YOLOv5s and DeepSORT algorithm to automatically detect and track the four behaviors of group-housed pigs under commercial conditions. The four class behaviors included lie, eat, stand and attack.

The detection model firstly adopts the improved YOLOv5s which combined the C3CBAM model to handle the constantly changing farm conditions, e.g., lighting conditions, problems of occlusion caused by other pigs. Then, to improves the robustness of the model, we adopt the Shape-IoU [23] to optimize the bounding box regression loss function of YOLv5s. Finally, to obtain the better performance of tracking, the improved DeepSORT architecture then is utilized to monitor each pen area covering all pig and track each pig behavior changes. In this way, the proposed improved tracking system avoids pigs' ID error switches, which can improve the tracking equality due to the missed and false pig detections.

The main contributions of this study are shown as bellow:

(1) The improved YOLOv5s and DeepSORT approach is proposed for detection and tracking in the field of pig's behavior analysis.

(2) The detector adopts the improved YOLOv5s which combines the best attention mechanism model (C3CBAM) to enhance the detection performance.

(3) The Shape-IoU loss function is utilized on the YOLOv5s detector to cope with the missed and false pig detections caused by variable farm environment.

(4) We accomplish the detailed comparison experiments and provide the discussion in different lighting and density condition for pig detection and tracking.

## 2 Materials and methods

### 2.1 Materials

The experimental data includes two parts, one part comes from the data set in the literature [24] and other part data is collected by FL3-U3-88S2C-C camera in Sep 1–30, 2022, from Lejiazhuang Pig Farm in Foshan, China. Fig 1a shows a video surveillance system for data acquisition, with the camera installed at the center of the ceiling to ensure broad coverage of the pig group's activities. Fig 1b shows the collected image, which includes 16 pigs in one pen. Although the camera angle is fixed, this setup has proven effective in the context of this study, capturing the behavior of most pigs. In this study, 12 videos of pig behavior have been filmed and labeled. Videos of the pig behavior were recorded at 25 frames per second (FPS) with image frame width of 2688 pixels and frame height of 1520 pixels. The 2,400 image sequences are extracted from the 12 videos data and manual annotated for four behavior classes. Table 1 presents the definitions of four behavior types, including standing, lying, eating, and attacking, for pig detection and tracking. To ensure the adequacy and diversity of the data, the "attack" behavior category covers four subtypes: (1) two pigs making head contact and rapidly

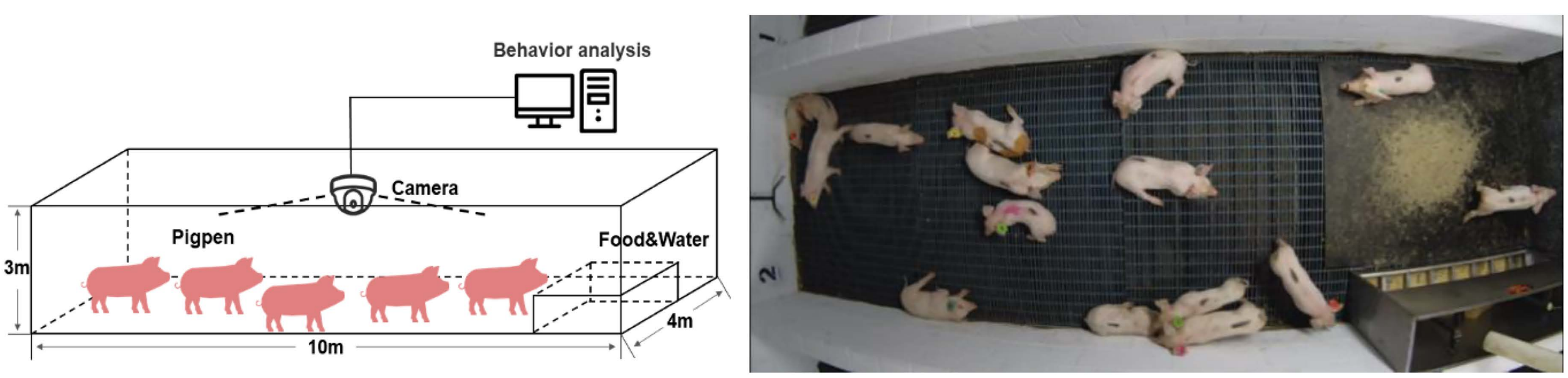


(a) (b)

**Fig 1. Data acquisition system.** (a) video surveillance system for data acquisition, (b) a collected image including 16 pigs for one pen.


Table 1. The definition of four types of behaviors and instances number after data augmentation.

| Pig Posture and Label | Identification Convention | Instances | Data Augmentation |
|---|---|---|---|
| Stand | Only feet or feet and snout in contact with the floor | 14945 | 17934 |
| Lie | Belly and folded limbs in contact with the floor | 11906 | 14287 |
| Eat | Only head in contact with the food and water trough | 1905 | 2286 |
| Attack | Two pigs with head contact and rapid head bobbing | 65 pairs | 1424 pairs |
| | One pig head contact with the body of the other pig | 85 pairs | |
| | Two pigs are moving rapidly towards each other | 575 pairs | |
| | One pig bites the other pig's tail | 224 pairs | |


bobbing their heads (65 pairs), (2) one pig making head contact with the body of another pig (85 pairs), (3) two pigs rapidly moving toward each other (575 pairs), and (4) one pig biting the tail of another pig (224 pairs).

Among the dataset, a total of 35931 instances after data augmentation operation with four different types of behaviors is used for training YOLOv5s detector according to 7:3 ratio of training and test sets. The animal study protocol was approved by the Animal Ethics Committee of South China Agricultural University (protocol code 2024F213 and date of approval:14 March 2024).

During the tracking procedure, there are 12 videos as shown in Table 2, which depict pigs' spare/dense status, number of pigs, variations in housing facilities, basic activity levels, and lighting scenarios. The dataset is annotated using DarkLabel software for training the DeepSORT model. And 8 videos are selected as train set, other 4 videos (Pig02, Pig06, Pig09 and Pig13) are used for testing model. The training and test samples contain all typical and challenging cases for pig images processing.

### 2.2 Methods

The method used in this paper is shown in Fig 2. The pipeline is divided into two parts. Pig behavior recognition is detected via an improved YOLOv5s network and the pigs with these behaviors are tracked using improved DeepSORT model. Once the pigs with four types of behaviors are detected by YOLOv5s, the positions and behavior classes of the targets during the current frame are transferred to the improved DeepSORT tracker, then the targets in the subsequent frames are tracked with the unique ID number. The target detection algorithm continues to detect subsequent frames, and the detected targets are passed to the tracking algorithm to determine whether they are the tracked targets.

**2.2.1 The proposed improved YOLOv5s detector.** The detector is YOLO (You Only Look Once) model based on the current classical one-stage algorithm (Redmon et al., 2016). There are four types of YOLO models (YOLOv5s, YOLOv5m, YOLOv5l and YOLOv5x), depending on the depth and width of the network. Among them, YOLOv5s is a lighter and faster version of all the YOLO algorithms, so it is a suitable framework for the detection of the household-raised pig. Its main structure includes input, backbone, neck and head. The input of YOLOv5s plays the role of resizing

Table 2. Properties of 12 videos annotated for pig behavior tracking performance analysis.

| Video# | Pig01 | Pig02 | Pig03 | Pig04 | Pig05 | Pig06 | Pig07 | Pig08 | Pig09 | Pig11 | Pig12 | Pig13 |
|---|---|---|---|---|---|---|---|---|---|---|---|---|
| Day | √ | √ | | √ | √ | | √ | | √ | √ | √ | |
| Night | | | √ | | | √ | | √ | | | | √ |
| #of pigs | 7 | 7 | 15 | 16 | 16 | 14 | 16 | 16 | 16 | 12 | 15 | 16 |
| Activity level | H | H | L | M | H | H | H | H | H | M | H | H |
| Sparse/Dense | Sparse | Sparse | Dense | Dense | Dense | Dense | Dense | Dense | Dense | Dense | Dense | Dense |
| Train/Test | Train | Test | Train | Train | Train | Test | Train | Train | Test | Train | Train | Test |


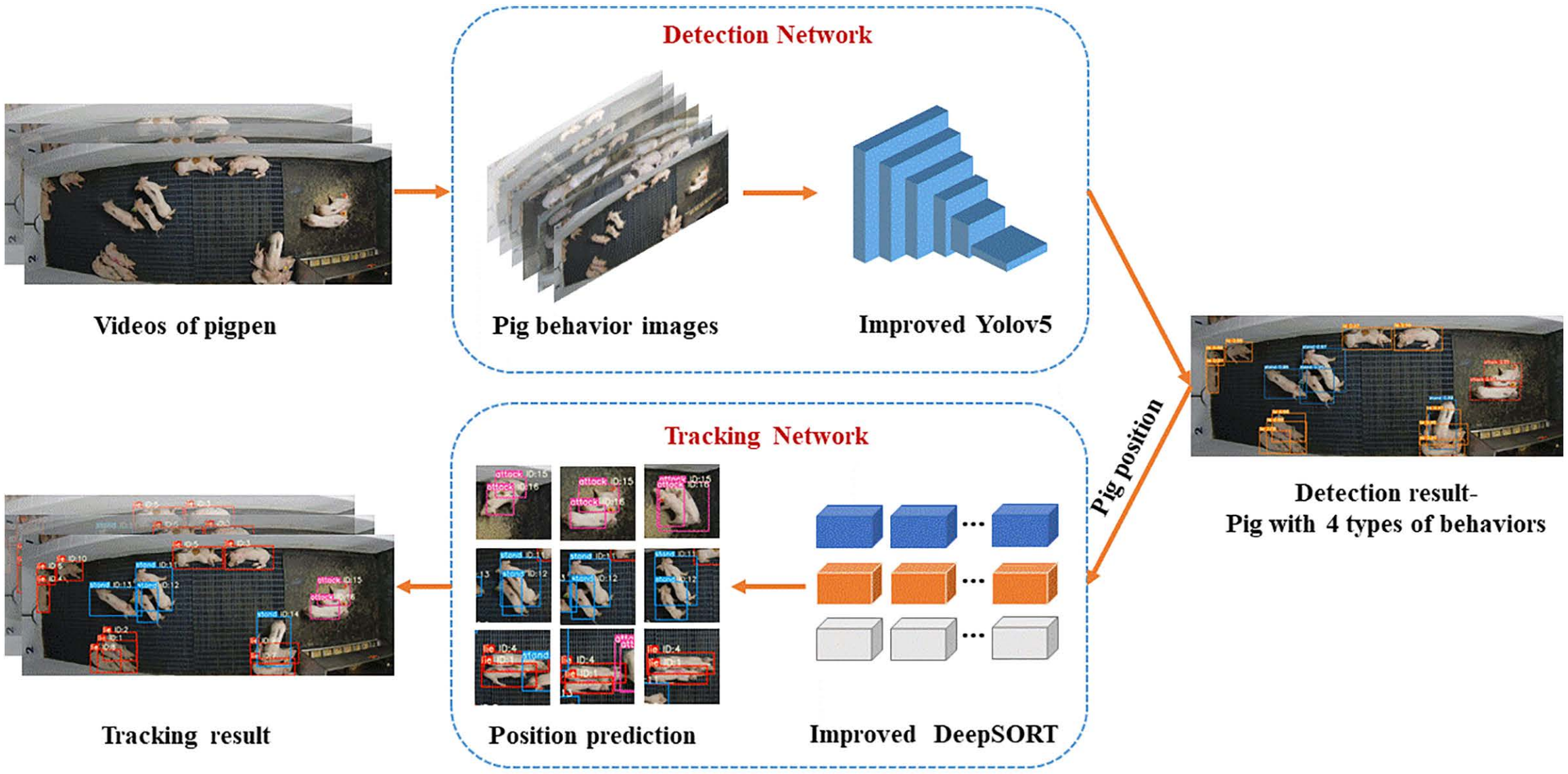


**Fig 2. The improved YOLOv5s and DeepSORT framework diagram.**


the video stream image received and performing Mosaic data augmentation, the C3 in backbone of YOLOv5s are used to extract the feature information, its neck is made of FPN and PAN which are used to fuse the rich features extracted by the backbone part. YOLOv5s has the relative simpler structure and runs fast, which meets the requirement of real-time detection. However, the detection precision of YOLOv5s will decrease when the targets are dense or occluded under farm demanding conditions, e.g., illumination conditions and occlusion.

In order to address this issue, this study integrates the CBAM (Channel and Spatial Attention Module) and Shape-IoU loss function into YOLOv5s to enhance its performance in pig behavior detection. Although CBAM and Shape-IoU have been widely used in other domains (such as image classification and object detection), their application in pig behavior detection offers unique advantages. Compared to other attention mechanisms, CBAM effectively focuses on the key information in images, especially in farming environments where lighting variations and occlusions significantly impact detection accuracy. By adaptively adjusting channel and spatial attention, CBAM helps the model better understand the target features in complex backgrounds, thus improving detection accuracy in complex scenarios.

Additionally, Shape-IoU, as a novel loss function, provides more precise optimization of the target bounding box shape compared to traditional IoU metrics (such as CIoU and DIoU). In pig farming scenarios, the posture and spatial variations of pigs are complex, and traditional IoU loss may not adequately handle these variations. Shape-IoU, by more accurately calculating the overlap of target shapes, effectively improves the regression accuracy of the detection box, especially when targets are occluded or partially overlapping.

Therefore, by integrating CBAM and Shape-IoU into YOLOv5s, this study not only improves the model's detection accuracy in dense farming environments but also maintains high detection stability under complex conditions such as lighting changes and target occlusions.

The improvements of YOLOv5s algorithm in Fig 2 are shown as follows:

(1) The improved CBAM and C3CBAM modules in the backbone of YOLOv5s model were proposed for the detection performance enhancement.

(2) The DIoU_Loss function of the head in original YOLOv5s is displaced by Shape-IoU to obtain a better convergence effect.

CBAM (shown in Fig 3) includes CAM (Channel Attention Module) and SAM (Spartial Attention Module) parts used for channel and space attention. This paper proposes two ways to add CBAM modules. The one way is to insert CBAM module ahead of C6 component, and other way is to insert CBAM modules in all C3 of Backbone, which is called C3CBAM. This study did a lot of experiments to compare the effect between the CBAM and the C3CBAM (shown in Table 3). The C3

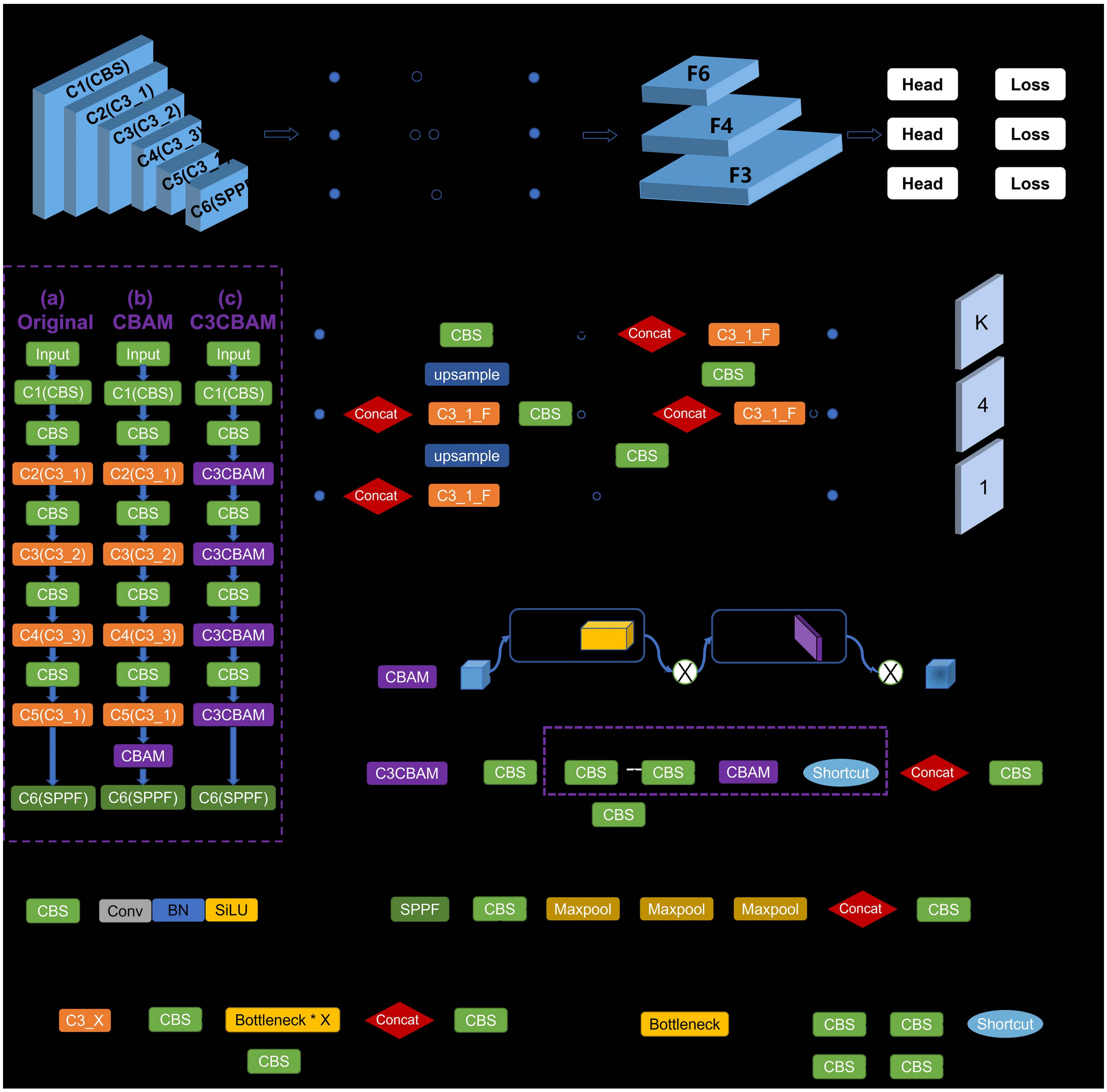


**Fig 3. YOLOv5s model with attention modules.**


**Table 3. Comparison of YOLOv5s with different strategy methods.**

| CBAM | C3CBAM | Shape-IoU | Precision↑ | Recall↑ | mAP@0.5↑ | FPS | GFLOPs |
|---|---|---|---|---|---|---|---|
| – | – | – | 94.7 | 96.6 | 96.2 | 43.8 | 15.8 |
| √ | – | – | 95.7 | 97.1 | 97.3 | 41.1 | 16.1 |
| – | √ | – | 97.3 | 98.5 | 98.7 | 40.9 | 16.0 |
| √ | – | √ | 96.1 | 97.5 | 97.9 | 41.1 | 16.1 |
| – | √ | √ | 98.2 | 99.2 | 99.1 | 40.9 | 16.0 |

mAP@0.5 means the mean average precision when IoU value equals to 0.5. The best results are shown in **bold**.


module is actually a simplified version of bottleneck CSP (Center and Scale Prediction). In addition to bottleneck, this part has only 3 convolution modules, which can reduce parameters, so it is named C3. The X in C3_X refers to the number of bottleneck modules included by the C3 module. For example, C3_1 is a C3 module containing a single bottleneck. The advantage of C3 is to simplify the network structure, reduce the amount of calculation and model inference time, and the performance of the model does not decline. In the combination method of C3 and CBAM, the CBAM is used to replace the bottleneck of the C3. Combination method is shown in the purple dashed part of C3CBAM on Fig 3.

Using Shape-IoU as the loss function has faster and better convergence effect, it can adapt to detect objects under situation of the high density of group pigs. The penalty term for Shape-IoU is shown in Eq. 1.

$$\mathrm{L}_{\mathrm{Shape-IoU}} = 1 - IoU + distance^{shape} + \Omega^{shape}/2 \tag{1}$$

$$IoU = \frac{\left|\mathrm{B} \cap \mathrm{B}^{\mathrm{gt}}\right|}{\left|\mathrm{B} \cup \mathrm{B}^{\mathrm{gt}}\right|} \tag{2}$$

$$distance^{shape} = hh \times \left(\frac{x_c - x_c^{gt}}{c}\right)^2 + ww \times \left(\frac{y_c - y_c^{gt}}{c}\right)^2 \tag{3}$$

$$hh = \frac{2 \times \left(w^{\mathrm{gt}}\right)^{\mathrm{scale}}}{\left(w^{\mathrm{gt}}\right)^{\mathrm{scale}} + \left(h^{\mathrm{gt}}\right)^{\mathrm{scale}}} \qquad hh = \frac{2 \times \left(h^{\mathrm{gt}}\right)^{\mathrm{scale}}}{\left(w^{\mathrm{gt}}\right)^{\mathrm{scale}} + \left(h^{\mathrm{gt}}\right)^{\mathrm{scale}}} \tag{4}$$

where B and $\mathrm{B}^{\mathrm{gt}}$ represent the predicted box and the GT box, respectively. $(x_c, y_c)$ and $(x_c^{gt}, y_c^{gt})$ are the center points of anchor box and GT box respectively. $w^{gt}$ and $h^{gt}$ represent the width and height of real frame, respectively. scale is the scale factor, which is related to the scale of the target in the dataset, and *ww* and *hh* are the weight coefficients in the horizontal and vertical directions respectively, whose values are related to the shape of the GT box. $\Omega^{shape}$ is shown in Eq. 5

$$\Omega^{shape} = \sum_{t=w,h} \left(1 - e^{-w_t}\right)^{\theta}, \theta = 4 \tag{5}$$

$$\begin{cases} w_w = hh \times \frac{\left|w - w^{gt}\right|}{\max(w, w^{gt})} \\ w_h = ww \times \frac{\left|h - h^{gt}\right|}{\max(h, h^{gt})} \end{cases} \tag{6}$$

Where $w$ and $h$ represent the width and height of predicted frame, respectively.

**2.2.2 The improved DeepSORT tracker.** The DeepSORT algorithm is an improved algorithm from SORT algorithm. The algorithm includes three parts, Hungarian matching algorithm, Kalman filter (KF) module, and track management.

In the group-housed pigs tracking application, as the video frames grow, DeepSORT will assign different ID values to the same pig target, resulting in the maximum ID value of the pig will significantly exceed the number of real pig targets. In addition, the pig IDs are changed wrongly in tracking, the main reason is that the detection results cannot match with the original tracks when the target pigs are moving or overlapped by occlusion, resulting in new tracks generated from unmatched detection results. According to the total number of the pigs in a single pigsty can be approximately seen as a constant (equal to n), we proposed an improved DeepSORT used to limit the target object ID error growth. Fig 4 shows the flow chart of the tracking process of the improved DeepSORT. Its process is list as follows:

(1) Cascade matching is used for the first matching of detections and confirmed tracks during consecutive frame sequences, and the matching tracks are updated with KF based on assigned detections.

(2) Unconfirmed tracks, unconfirmed tracks in cascade matching, and unmatched detections are matched for the IOU matching using the Hungarian assignment algorithm, the matching tracks are also updated with KF based on assigned

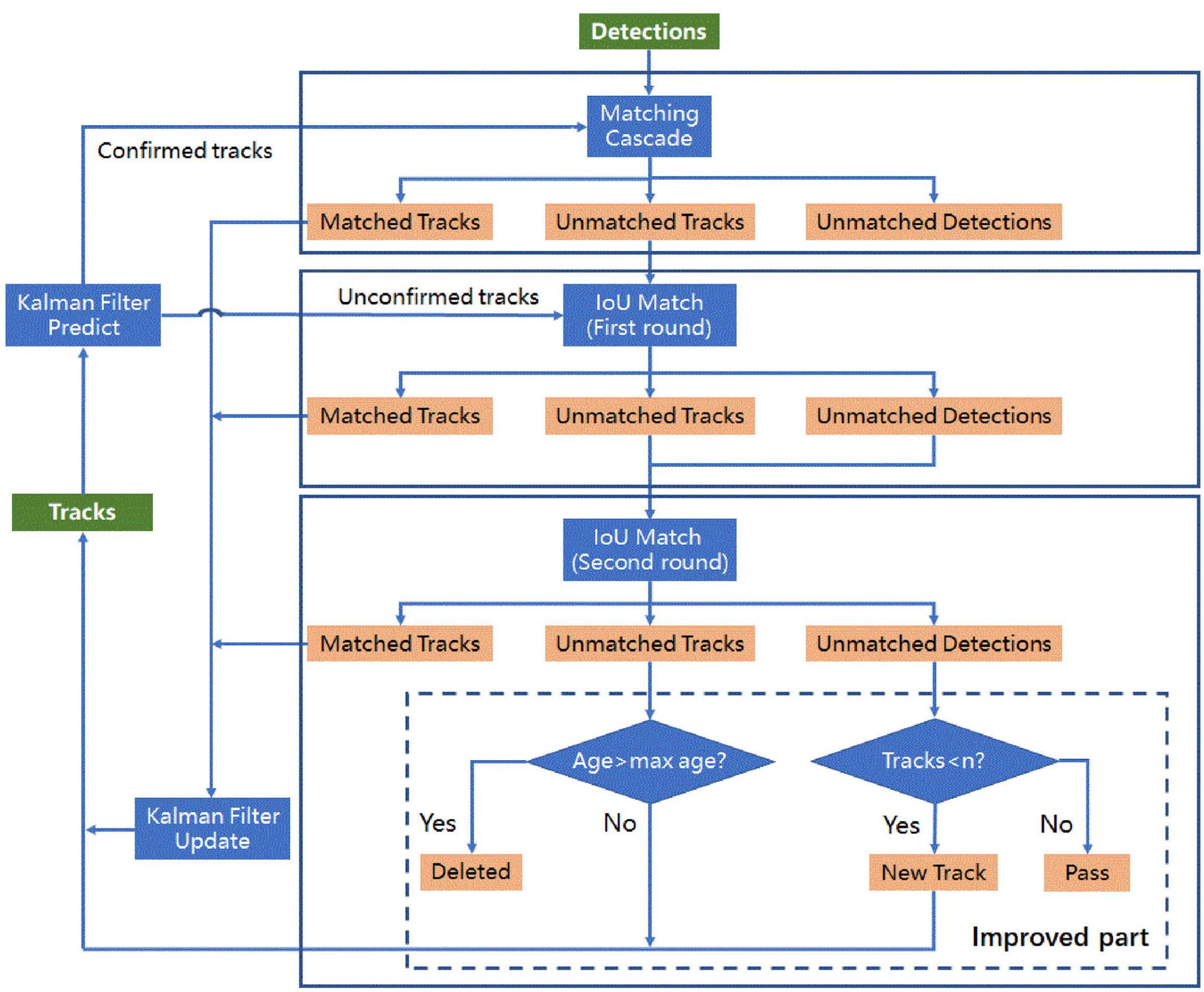


**Fig 4. The improved DeepSORT framework.**


detections. To further improve the matching accuracy, a second round of IOU matching is performed between the remaining unmatched tracks and unmatched detections. This step helps reduce ID switching caused by short-term occlusion or fast movements. The successfully matched tracks are also updated with KF.

(3) For unmatched detections that remain after the two rounds of IOU matching, if the image is the first frame, we created the new tracks, if not, then we create the new tracks when the condition is satisfied that $n >$ the count of tracks. The value of n is determined by the average of the number of detected targets in nearest 3 frames. For unmatched tracks, we deleted them when the age of tracks exceeds the max age (set as a maximum of 30 frames in this paper).

For piggery-specific scenarios, the improved DeepSORT algorithm limited the target object ID growth, this allows the maximum ID value of the pig will not exceed the number of real pig targets. And it can improve track generation and deletion in the tracking process.

## 3 Result

### 3.1 Experimental configuration

In this study, YOLOv5s+DeepSORT are built on Pytorch and trained on RTX2070 GPU with a total of 8 images per mini batch. The operating system is Windows 10, and we adopt Adam optimizer with a small initial learning rate $5 \times 10^{-5}$ with weight decay 0.0001. For the detection model, 2,400 images of pig's behavior data are used totally for training. Considering the GPU memory limitation, the batch-size is set to 8 during training while a total of 300 iterations is performed. The tracking model is trained with 8 videos and 37,500 annotated images. The essence of training tracking model is to train the Re-ID CNN of the DeepSORT.

### 3.2 Results of YOLOv5s improvement

**3.2.1 Results of attention improvement on YOLOv5s.** This study evaluates the effects of different improvement strategies of the YOLOv5s mentioned previously. Comparative experiments are carried out on the same test data. CBAM, C3CBAM, and CIoU Loss are added to the initial YOLOv5s for comparison. This study uses AP@0.5, Precision and Recall as detection evaluation criteria to compare the effect of different improving strategies.

The results are shown in Table 3. Note that the five approaches are all built on YOLOv5s. We can see that the value of mAP@0.5 only using CBAM increases 1.1% compared with not using CBAM, which indicates that the adopting attention module can get a better performance. C3CBAM can further improve the mAP@0.5 to 98.7%. According to experimental results (Table 3 the second and third row), CBAM and C3CBAM have improved the detection results. In addition, Shape-IoU function combined by the attention module has improved performance slightly weaker than that only using attention mechanism. The attention mechanism aims to improve the network's ability to extract important features, which is the key to improvement in performance, while Shape-IoU contributes to speeding up the regression of the prediction frame. The improvement effect of adopting the Shape-IoU function also is proved by the results, where the mAP@0.5 of C3CBAM+Shape-IoU increased by 0.4% comparing to the C3BAM one. The results of the five methods are similar in terms of computing speed FPS and GFLOPs metrics, and there is no significant difference in speed. As the strategy using the C3CBAM+Shape-IoU gets the best mAP@0.5 with 99.1%, we use YOLOv5s with C3CBAM+Shape-IoU method for target detection in the rest of our experiments.

**3.2.2 Behaviors recognition results of YOLv5s and improved YOLOv5s.** We compare the YOLOv5s and the improved one using the PR (Precision-Recall) curves. The PR curves of four different behaviors are shown in Fig 5. In PR curve, the curve is closer to the upper right corner, the better is the performance. The improved model performs better than that of the YOLOv5s. We can see that the mAP@0.5 of all labels are more than 90% in original YOLOv5s (Fig 5, Left). Of the four types of behaviors recognition, the attack behavior achieves the worst result of mAP@0.5 with 93.1%. The PR curve of the improved YOLOv5s (Fig 5, right) shows that the mAP@0.5 of all behavior reach to 99.1% and the mAP@0.5 of

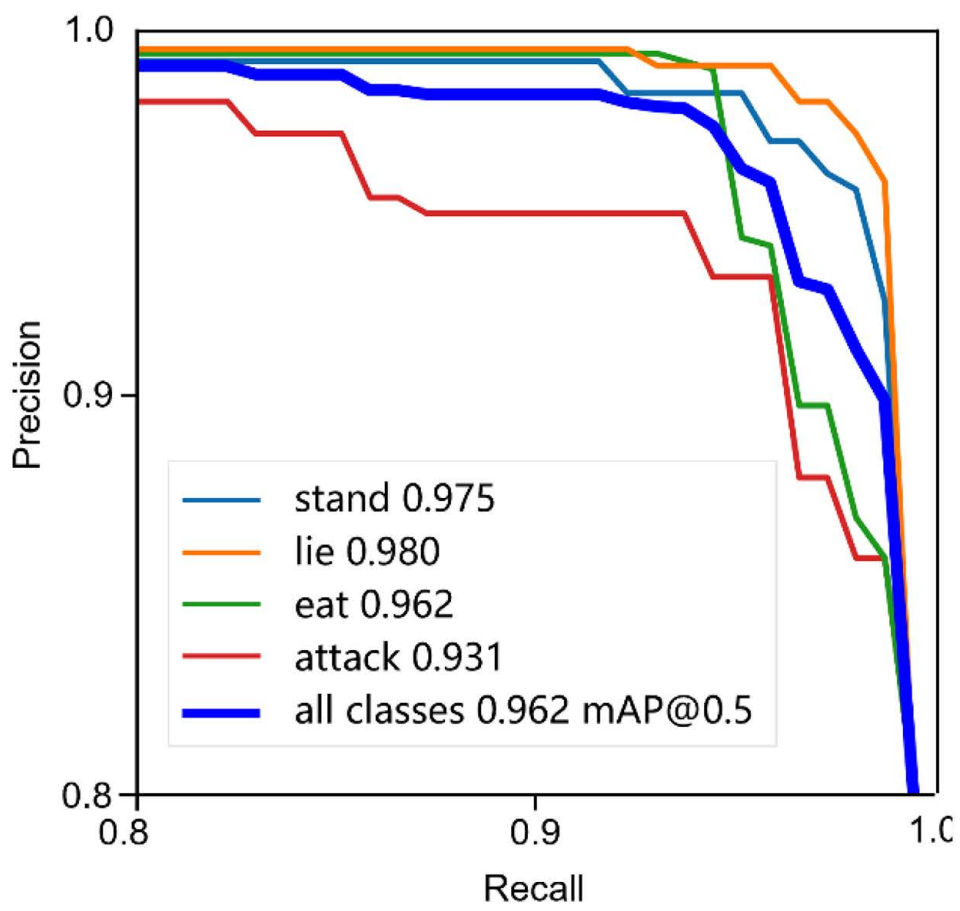


(**a**) original YOLOv5s

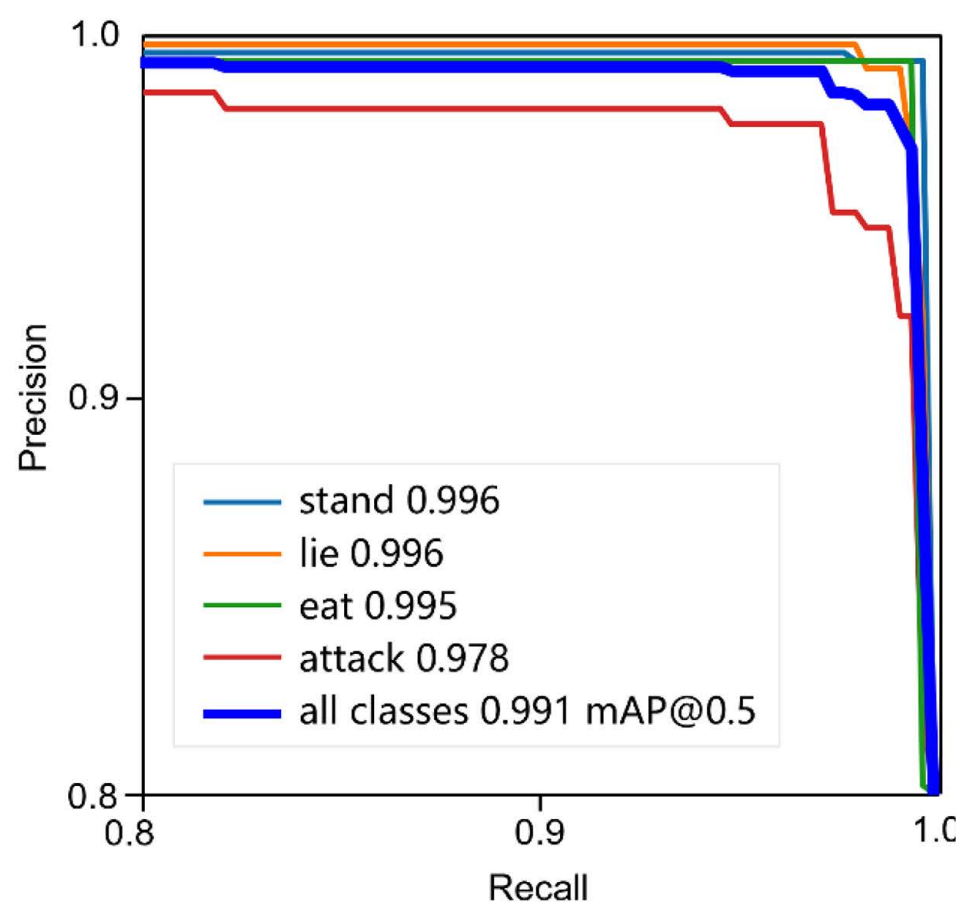


(**b**) improved YOLOv5s

**Fig 5. PR curves of improved YOLOv5s.**


attack is up to 97.8%, raise by 4.7%. The mAP@0.5 of all kinds of behaviors on the improved algorithm increase by 2.9%. It proves that the improved YOLOv5s using C3CBAM+Shape-IoU is effective for improving the detector performance.

We evaluate the performance of improved YOLOv5s on detecting 4 kinds of behaviors, the comparative results between improved YOLOv5s and the original one, using precision, recall, F1, AP@0.5 and mAP@0.5 as evaluation metrics are given. The comparison results are shown at Table 4. The improved YOLOv5s achieves better performance than the one on all the metrics. The improved YOLOv5s achieves 99.1% with mAP@0.5 and 98.7% with F1 value, 2.9% higher and 3.1% than YOLOv5s method, respectively. For both approaches, we can find that the F1 score and AP@0.5 of lie behavior is the highest one of all the labels, followed by the stand and eat behavior. It is related to the difficulty of detecting the frequently changing behavior. In addition, attack behavior recognition achieves the worst result on all the evaluation metrics. This is mainly because, unlike other behaviors, the attack behavior changes quickly and is interactive between multiple pigs. All these results suggest the performance of improved YOLOv5s on classifying four kinds of behaviors is better than the original one.

Fig 6 presents the visualization of the detection results of YOLOv5s and the improved method on the test set. From the results in Fig 6b, it can be seen that our method, using the C3CBAM attention module, effectively addresses the issues of

**Table 4. Comparison of original YOLOv5s and improved YOLOv5s.**

| Algorithm | Behaviors | P↑ | R↑ | F1↑ | AP@0.5↑ | mAP@0.5↑ |
|---|---|---|---|---|---|---|
| YOLOv5s | stand | 95.6 | 97.2 | 96.4 | 97.5 | – |
| | lie | 95.7 | 97.3 | 96.5 | 98.0 | – |
| | eat | 94.8 | 96.9 | 95.8 | 96.2 | – |
| | attack | 92.7 | 95.0 | 93.8 | 93.1 | – |
| | all | **94.7** | **96.6** | **95.6** | **–** | **96.2** |
| Improved YOLOv5s | stand | 99.4 | 99.8 | 99.6 | 99.6 | – |
| | lie | 99.6 | 99.9 | 99.7 | 99.6 | – |
| | eat | 98.8 | 99.2 | 99.0 | 99.5 | – |
| | attack | 95.1 | 97.9 | 96.5 | 97.8 | – |
| | all | **98.2** | **99.2** | **98.7** | **–** | **99.1** |


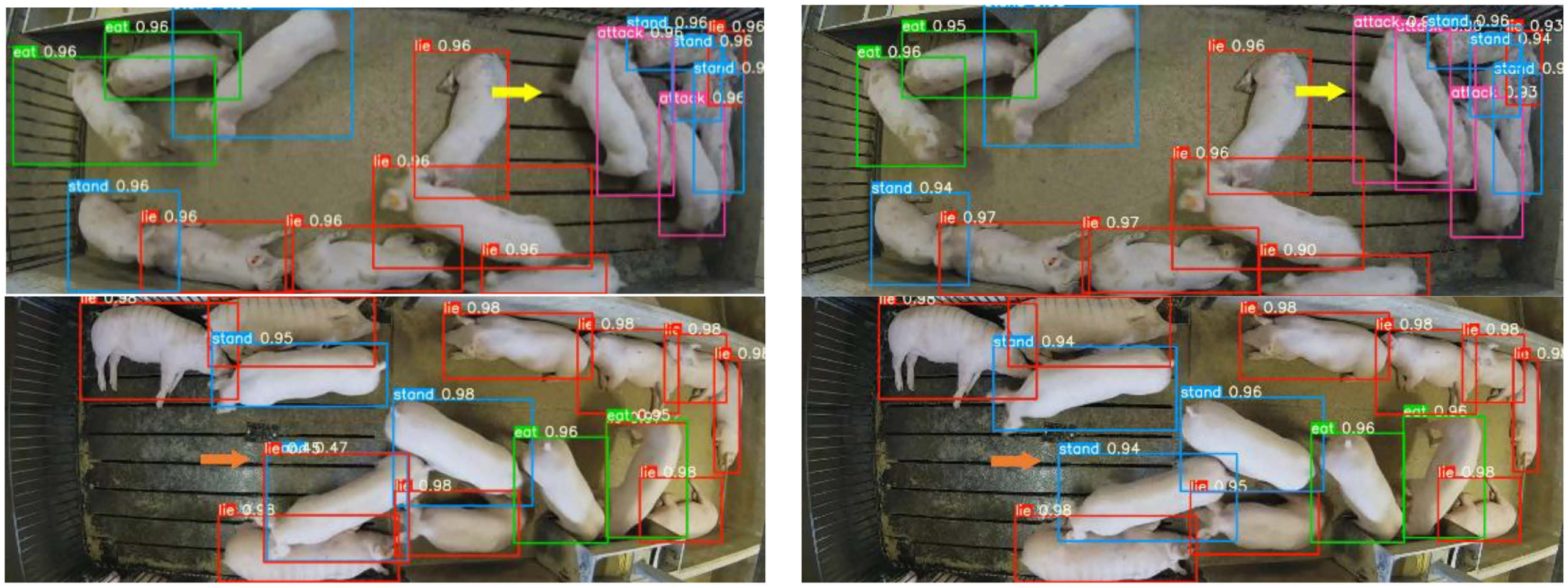

(a) YOLOv5s (b) Improved YOLOv5s

**Fig 6. Detection results between YOLOv5s and improved YOLOv5s.** (a)YOLOv5s, (b) Improved YOLOv5s.


false and missed detections when pigs are clustered together, as indicated by the orange and yellow arrows, respectively. According to the experimental results (as shown in the first and second rows of Fig 6a with yellow and orange arrows), YOLOv5s detection fails to recognize the attack behavior due to missed detection when pigs are clustered together, while our improved method successfully detects the behavior (yellow arrow). In the second row, YOLOv5s incorrectly detects a standing pig as lying down, while our method correctly detects the standing pig (orange arrow). This improvement is mainly attributed to the application of the C3CBAM attention mechanism.

Fig 7 illustrates several detection results of the improved YOLOv5s on the test set in different scenarios. From the results in Fig 7b, with the use of the C3CBAM attention mechanism, our method effectively overcomes false detection and missed detection issues. In the first row of Fig 7a, as indicated by the orange arrow, the YOLOv5s detector causes false

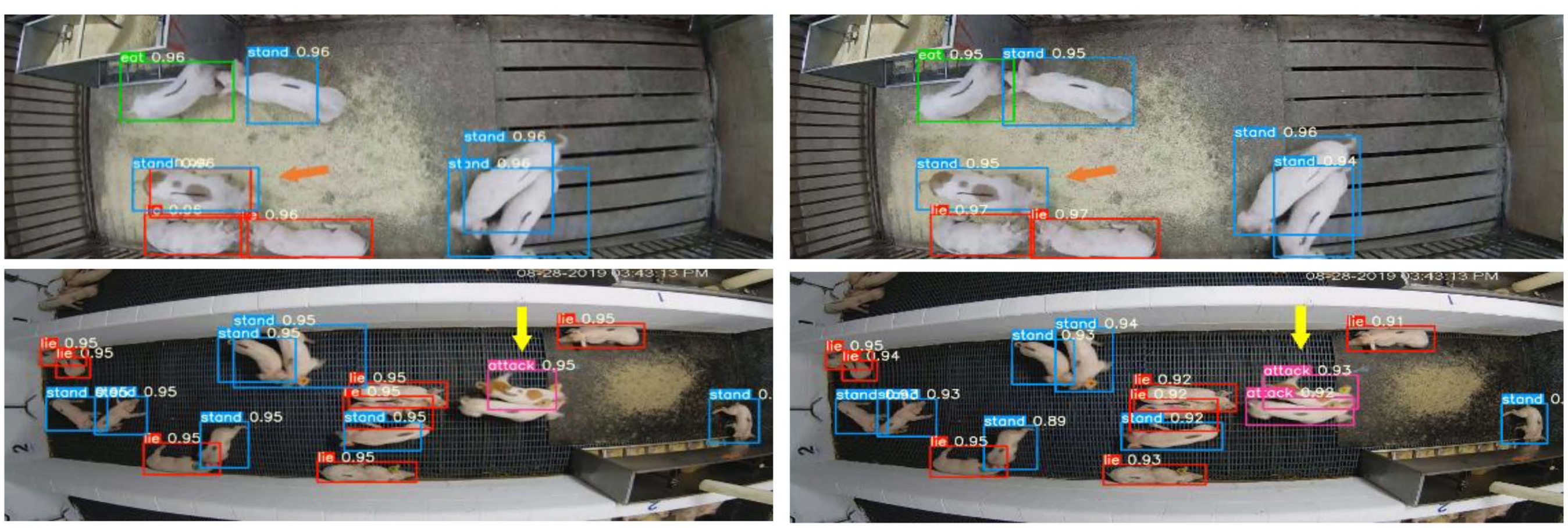

(a) (b)

**Fig 7. Example detection results between YOLOv5s and improved YOLOv5s under different scenes.** (a)YOLOv5s, (b)Improved YOLOv5s.


detection when the number of pigs is low. In the second row of Fig 7a, as indicated by the yellow arrow, the YOLOv5s detector suffers from missed detection when the number of pigs is high. In contrast, the improved method maintains high detection accuracy across different scenarios, whether there are fewer or more pigs, and effectively avoids both false and missed detections. Therefore, the improved method, based on the C3CBAM attention mechanism, provides more stable and precise detection results in complex scenarios, further enhancing the robustness and practicality of the model in dynamic environments.

### 3.3 Tracking result of improved DeepSORT tracker

The improved DeepSORT is compared with original DeepSORT using the same detector in group-housed pigs tracking experiment. Table 5 shows the performance of improved DeepSORT and the original DeepSORT in the test data.

From Table 5, the improved DeepSORT achieves 94.5% of MOTA,94.9% of MOTP and 72.8 of IDF1 respectively. Compared with DeepSORT, the results of its MOTA, MOTP and IDF1 in 4 test videos increases by 8.6%, 8.2% and 3.7%, which improves from 85.9%,86.7% and 69.1% to 94.5%,94.9%, and 72.8% respectively. In all videos frames, the number of ID switchs (IDs) of the improved DeepSORT are less than that of DeepSORT model, which decreases the number of IDs from 299 to 216, decreasing by 12.7%. The results validate that the tracks generation and deletion module of improved DeepSORT brings targets tracking ability improvements and decreases the number of IDs. Based on the experimental results above, our method can be applied to the tracking of herd-raised pigs' behavior, which improves the tracking stability under the different environments.

Fig 8 visualizes several comparison tracking results between improved DeepSORT and DeepSORT on Pig02, Pig09 and Pig13 test videos. From the results of Fig 8 (the first row as shown orange arrows), we can see that our method can overcome the problem of false tracking, when the low-density pigs close to other pigs on the daytime. According to experimental results (Fig 8 the second row as shown yellow and orange arrows), the DeepSORT detector causes missed and false tracking results under circumstances of high-density on the daytime. However, the improved DeepSORT performs well under crowded scenes. And it can keep both correct bounding boxes and correct tracking when the pigs are heavily occluded at night (as shown in the Fig 8 the third-row yellow arrows).

Fig 9 visualizes several tracking results of improved DeepSORT on different conditions. From the results of Fig 9 (the first column), we can see that our method can overcome the problem of ID frequently switching, even when the pig of ID 6 and ID7 were fiercely fighting from the bottom right (frame 70) to the top left (frame 120). According to experimental results (Fig 9 the second column), our method performs well, and the ID keeps stable under crowded scenes at night. As the Fig 9 shown (the third column), it can keep correct tracking steadily and prevent the ID switch from remarkably increasing, even when there are pigs fighting on the top right. This mainly attributes to the improvement of the DeepSORT.

**Table 5. Comparision of DeepSORT and improved DeepSORT tracking model.**

| Algorithm | Test set | IDF1↑ | Rcll↑ | Prcn↑ | IDs↓ | MOTA↑ | MOTP↑ |
|---|---|---|---|---|---|---|---|
| DeepSORT<br>Improved DeepSORT | Pig02 | 75.5 | 95.1 | 97.8 | 30 | 93.3 | 96.2 |
| | Pig06 | 59.5 | 91.7 | 93.5 | 122 | 75.8 | 76.1 |
| | Pig13 | 83.9 | 95.7 | 96.4 | 55 | 90.5 | 88.2 |
| | Pig09 | 57.3 | 87.4 | 96.4 | 92 | 84.3 | 86.4 |
| | Total/average | **69.1** | **92.5** | **96.0** | **299** | **85.9** | **86.7** |
| | Pig02 | 82.5 | 95.5 | 98.8 | 19 | 94.0 | 96.4 |
| | Pig06 | 65.3 | 95.1 | 97.4 | 86 | 91.4 | 94.1 |
| | Pig13 | 80.1 | 98.9 | 99.3 | 35 | 97.5 | 95.2 |
| | Pig09 | 63.4 | 98.3 | 98.3 | 76 | 95.0 | 93.8 |
| | Total/average | **72.8** | **96.9** | **98.5** | **216** | **94.5** | **94.9** |


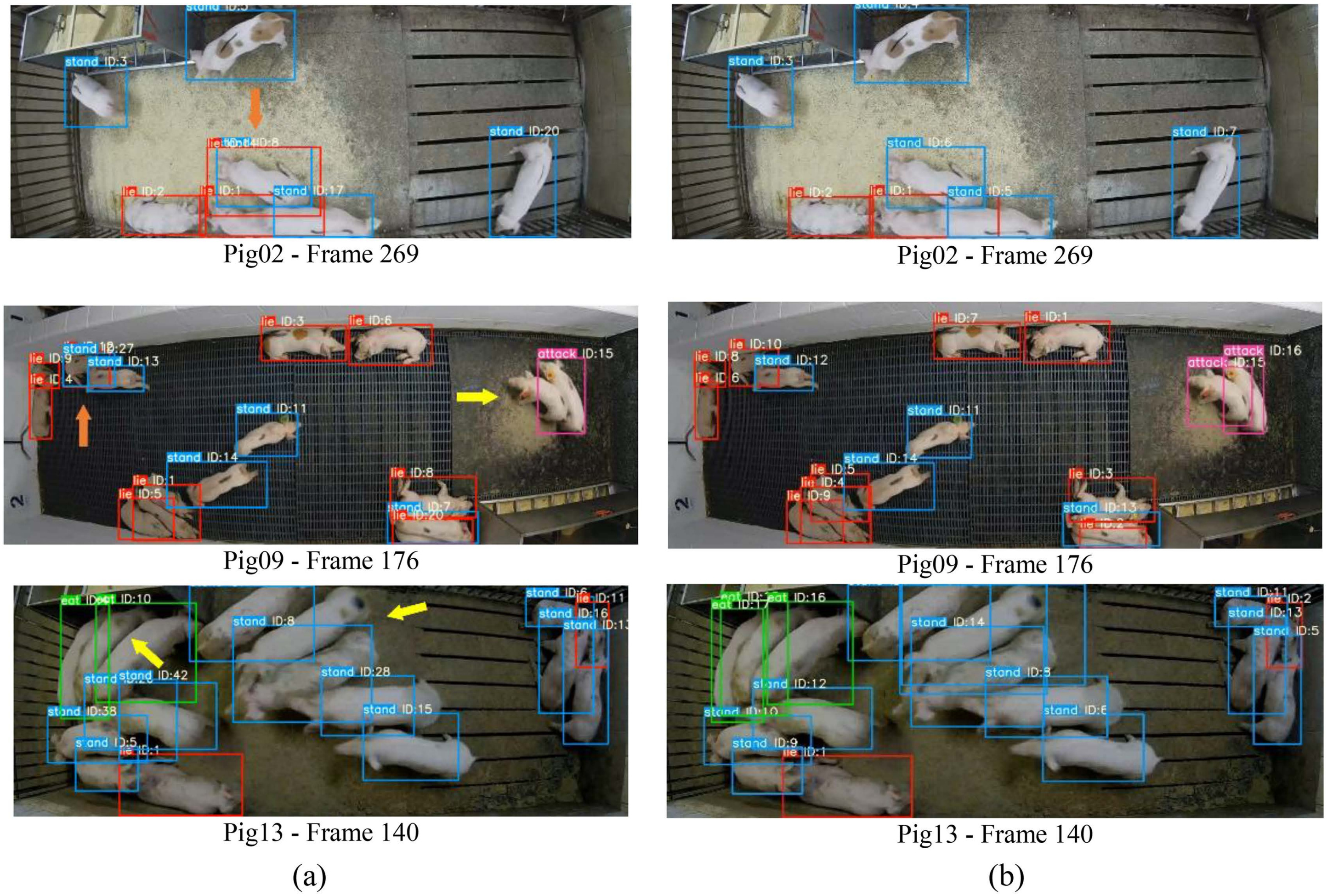


**Fig 8. Examples tracking results between DeepSORT and improved DeepSORT under different circumstances, (a)DeepSORT, (b) The improved DeepSORT.**


Fig 10 shows the comparison results between DeepSORT of improved DeepSORT model on occlusion conditions for the Pig09 video. The values of ID6 (shown in green arrows) are correct when 20 frames from the Pig09 video are not occluded using the unimproved and improved DeepSORT methods, as shown in the first line of Fig 10. However, at 113 frames, using the unimproved DeepSORT, the values of ID6 under occlusion mistakenly become 22 (shown in red arrows); and until 150 frames, the value of ID6 mistakenly switch to 22 (shown in red arrows). Using the improved DeepSORT method, no ID error switch has occurred, as shown in the second and third lines of Fig 10.

## 4 Discussion

In pig behaviors recognition task, target detection models play a key role in the performance of detecting 4 kinds of behaviors. There are significant differences due to farm demanding conditions, e.g., illumination conditions and occlusion of one pig from another. Therefore, in situations where it is not determined which target detector model is the most suitable, it is generally experimentally validated to choose the most appropriate target detector. To select the most suitable detector, this study also compared the results of the chosen improved YOLOv5s detector with those of the YOLOv5s, YOLOv8n, YOLOv9s, and YOLOv10n detectors, and the results are shown in Table 6. Compared with the other methods, the

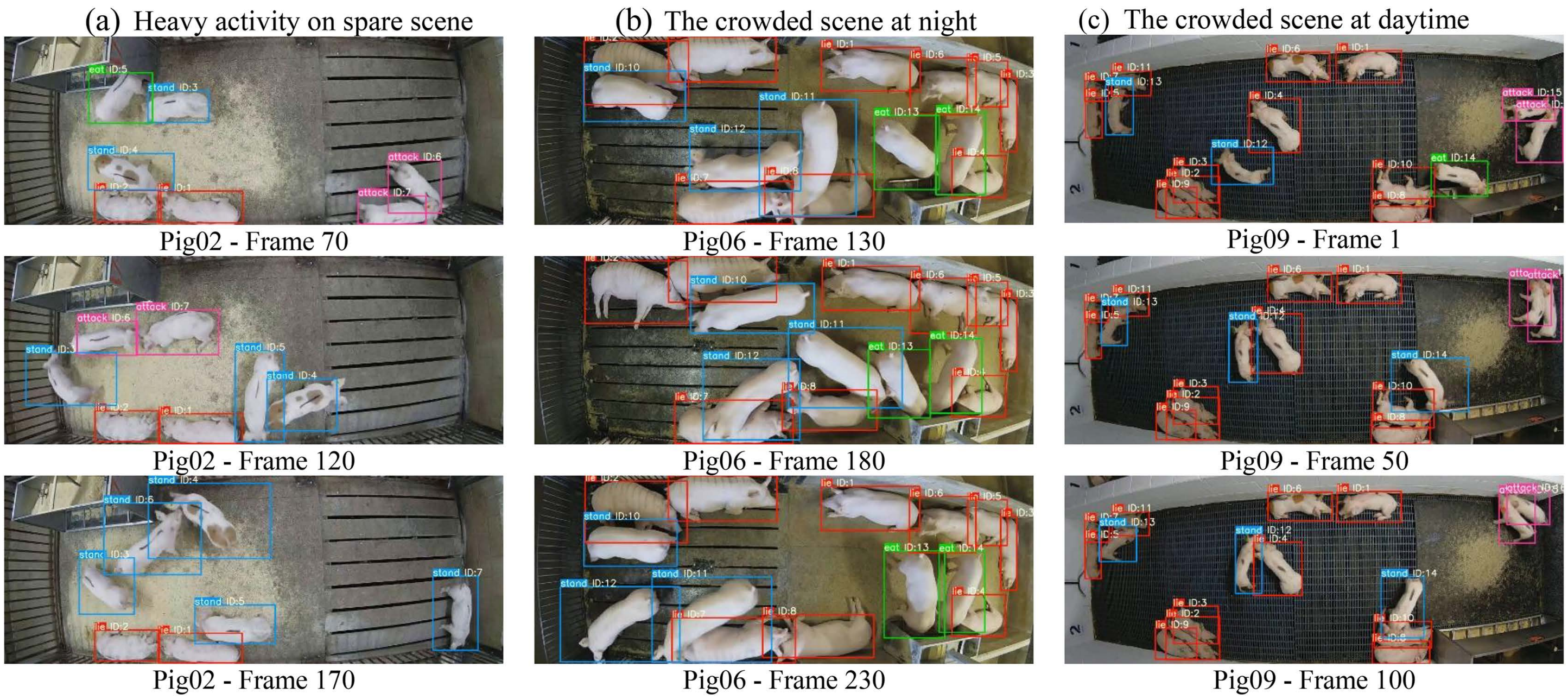


**Fig 9. Comparison diagram of improved DeepSORT model on different conditions.**


improved YOLOv5s achieves the best performance on all metrics. These comparative results further validate the effectiveness of improved YOLOv5s in pig detection and improving the accuracy of behaviors recognition.

This study also attempts to use the improved YOLOv5s and DeepSORT to analysis the behavior of household-raised pig. A series of results in this paper show that the improved algorithm has the advantages of detection accuracy and tracking stability. The improved algorithm can overcome the missed detection and false detection and the tracking unstable problem due to the complex environment and the frequently changes of pigs.

Table 7 presents a comparison of tracking results for improved DeepSORT, ByteTrack, OC-SORT, and DeepSORT. It is observed that improved DeepSORT outperforms other tracking algorithms in all metrics. Compared to ByteTrack, OC-SORT, and DeepSORT, our method achieves improvements in MOTA by 4.6%, 3.4%, 9.4%, in IDF1 by 2.9%, 2.3%, 3.7%, and in MOTP by 7.3%, 2.6%, 8.2%, respectively. Furthermore, our method shows the best performance in IDs with only 216. Overall, our method achieves the best results in IDF1, IDs, MOTA, and MOTP, which demonstrates the superior identity consistency of our approach, leading to more stable and reliable tracking performance.

Then, the high accuracy of the improved YOLOv5s and DeepSORT has been achieved in the detection and tracking experimental results. The mAP@0.5 of detection and the MOTP of tracking are 99.1% and 94.9% respectively, which is higher than the most of studies in this field. Secondly, the stability of the improved YOLOv5s and DeepSORT has been proved in the part of the comparison in different light and density condition, the IDs and MOTA do not drop significantly. This study utilizes the attention mechanism of C3BAM to improve the performance of the detector under different lighting and densities. The improved detector can significantly reduce missed and false detections. And the tracked ID has also become more stable.

The value of this study is that it successfully achieves the detection and tracking by using the architecture of improved YOLOv5s and DeepSORT and get a better accuracy and stability, which means it is closer to the application of industrial production. And the monitoring of pigs and doing the behavior analysis of the household-raised pig can prevent the pigs from hurting themselves and doing the other abnormal behavior that may have impact of the produce efficiency. This study can provide the reference algorithm for the industrial application of smart farm.

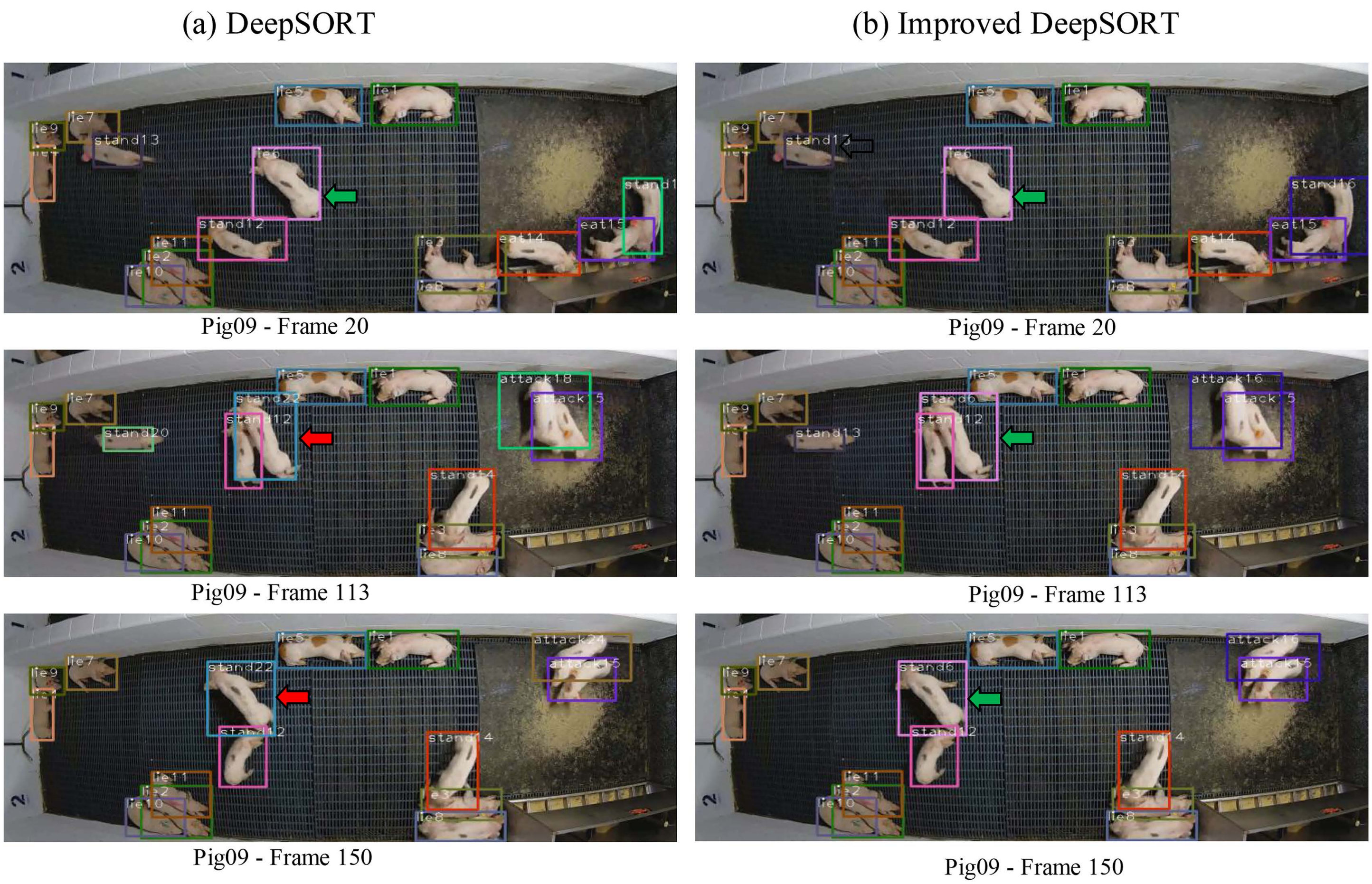


**Fig 10. Comparison results between DeepSORT of improved DeepSORT model on occlusion conditions.**


**Table 6. The comparison results of improved YOLOv5s with other detectors.**

| Detectors | P/%↑ | R/%↑ | mAP50/%↑ |
|---|---|---|---|
| YOLOv10n | 95.8 | 96.5 | 95.3 |
| YOLOv9s | 98.1 | 97.4 | 96.0 |
| YOLOv8n | 97.6 | 96.4 | 96.3 |
| Improve YOLOv5s | **98.2** | **99.2** | **99.1** |


**Table 7. The comparison results of improved DeepSORT with other trackers.**

| trackers | IDF1↑ | IDs↓ | MOTA↑ | MOTP↑ |
|---|---|---|---|---|
| ByteTrack | 69.9 | 278 | 89.9 | 87.6 |
| OC-SORT | 70.5 | 262 | 91.1 | 92.3 |
| DeepSORT | 69.1 | 299 | 85.9 | 86.7 |
| Improved DeepSORT | 72.8 | 216 | 94.5 | 94.9 |


## 5 Conclusions and future work

This study fuses the attention mechanism CBAM with YOLOv5s and adopts the Shape-IoU method to reduce the missed detection and false detection of target pigs due to complex problems such as occlusion and overlapping of pigs. The improved YOLOv5s can detect most of the pigs that are covered or occluded. Experimental results show that the mAP@0.5% of detection algorithm increase from 92.7% to 99.3%, which means 6.6% accuracy improvement. In terms of tracking. the evaluation values of MOTA and MOTP are 94.5% and 94.9% respectively.

And we carried out the comparison tests under different lighting and density conditions, where the experimental results prove the effectiveness and stability of the improved algorithm. so, the improved detection and tracking algorithm can meet the needs in the actual farming environment and provide technical support for contactless automatic monitoring of pigs, which has good engineering application prospects for the development of smart pig management.

## Author contributions

**Conceptualization:** Tianyu Cheng, Fujie Sun, Haoxuan Ou, Hairan Yang.

**Data curation:** Tianyu Cheng, Liang Mao, Haoxuan Ou, Shuqin Tu, Fang Yuan.

**Formal analysis:** Haoxuan Ou, Shuqin Tu.

**Funding acquisition:** Haoxuan Ou.

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
