## [Decision Letter · Decision Letter 0]

11 Apr 2025

Dear Dr. shuqin,

Thank you for submitting your manuscript to PLOS ONE. After careful consideration, we feel that it has merit but does not fully meet PLOS ONE’s publication criteria as it currently stands. Therefore, we invite you to submit a revised version of the manuscript that addresses the points raised during the review process.

We look forward to receiving your revised manuscript.

Kind regards,

Himadri Majumder, Ph.D

Academic Editor

PLOS ONE

Journal Requirements:

5. Please include a copy of Table 6 which you refer to in your text on page 15.

Reviewers' comments:

Reviewer's Responses to Questions

**Comments to the Author**

1. Is the manuscript technically sound, and do the data support the conclusions?

Reviewer #1: Partly

Reviewer #2: Partly

2. Has the statistical analysis been performed appropriately and rigorously?

Reviewer #1: Yes

Reviewer #2: Yes

3. Have the authors made all data underlying the findings in their manuscript fully available?

Reviewer #1: Yes

Reviewer #2: No

4. Is the manuscript presented in an intelligible fashion and written in standard English?

Reviewer #1: Yes

Reviewer #2: Yes

Reviewer #1: This paper proposes a method based on improved YOLOv5s and DeepSORT for attacking and daily behaviors detection and tracking of group-housed pig to enhance livestock production efficiency and animal welfare. Overall, the research content of this paper is practical. However, the effectiveness of the proposed method still requires more rigorous experimentation for validation. Additionally, the theoretical analysis in the experimental section needs to be further deepened and expanded to strengthen the scientific and persuasiveness of the study. Therefore, it is suggested that the author revise and improve the article before submitting it for review again.

1、In Figure 4, there are two “IOU Match” processes performed. However, the second matching process is not described in the text. Please carefully review the figure and text to ensure consistency.

2、In Figure 4, please clarify what parameter “age” refers to in the context. Additionally, the drawing logic of Figure 4 is not clear enough. It is suggested that the author redraw this figure.

3、In the section 2.2.2, the author aims to reduce the generation of new tracks by evaluating the relationship between “n” and “tracks”, where “n” is determined by the average of the number of detected objects in nearest 3 frames. However, the duration of three frames in a video is very short, possibly less than 0.2 seconds. In practical environments, the loss of tracking due to occlusion can last for several seconds or even minutes. Therefore, when occlusion exceeds three frames, “n” may be smaller than the actual number of pigs. When occluded pigs reappear, it remains unclear whether the proposed method can still achieve effective tracking. The author is recommended to include additional experiments or theoretical analyses at an appropriate location to validate the effectiveness of the method under occlusion conditions that exceed three frames.

4、The author demonstrates the effectiveness of the method in nighttime scenarios in Figure 6. The primary impact of nighttime conditions on model performance lies in the blurring of object features caused by low-light, which subsequently affects the recognition and tracking capabilities of model. However, in the experimental scenarios of Figure 6, the lighting conditions are relatively adequate and the object features are obvious. Therefore, the validity of using these scenarios to verify the model performance under low-light conditions is questionable. It is recommended that the author redesign this part of the experimental content to effectively validate the model performance in low-light environments.

5、In Figure 6, there are obvious errors in behavior recognition. For example, the “stand” is misidentified as “lie”. Moreover, the errors indicated by the yellow and orange arrows in the figure are clearly not caused by occlusion, which contradicts the analysis provided in the text. It is recommended that the author carefully verify the experimental results and conduct a correct theoretical analysis based on them to enhance the credibility of the article.

6、Figure 7 demonstrates the effectiveness of the model in crowded scenarios. However, the figure does not depict a high-density environment. In fact, the stocking density is even lower than that shown in Figure 6. Therefore, the effectiveness of the model in high-density environments remains to be verified. It is recommended that the author reselect test images and revise the corresponding experimental and analytical content.

7、In this study, the author has achieved the task of behavior classification of group-housed pigs through object detection. However, object detection only considers the spatial information of behaviors and lacks temporal information. The author also states in Table 1 that aggressive behavior is a dynamic process. So, how does the proposed method achieve accurate recognition without temporal information? For example, in Figure 7, two pigs with their heads close together were identified as standing. Similarly, another two pigs were identified as aggressive behavior. Why can the proposed method distinguish behaviors through the spatial relationships in the single frame image. Therefore, it is suggested that the author add a section in the introduction or experimental part to explain how the study ensures the accuracy of behavior detection in the absence of temporal information.

Reviewer #2: Below are my comments about the paper:

1.The integration of CBAM and Shape-IoU into YOLOv5s, while effective, is not fundamentally novel. Both CBAM (a well-established attention mechanism) and Shape-IoU (a recent but generic loss function) have been applied in other domains. The authors do not sufficiently justify why these components are uniquely suited to pig behavior detection compared to alternative attention mechanisms (e.g., SE, ECA) or IoU variants (e.g., CIoU, DIoU). The novelty lies primarily in their application to this specific use case, but this is not strongly differentiated from prior agricultural CV studies (e.g., reference 21 uses rotated bounding boxes, which might offer better occlusion handling).

2.While the improved YOLOv5s outperforms older YOLO versions (v8, v9, v10) in Table 6, this comparison lacks depth. For instance, ORP-Byte (reference 21) achieves MOTA of 99.8%, significantly higher than the 94.5% reported here. The authors should address why their method underperforms in tracking accuracy despite architectural improvements.

3.The ID-switch reduction mechanism in DeepSORT, which leverages the constant number of pigs per pen, is pragmatic but not groundbreaking. Similar constraints (e.g., fixed herd sizes) have been exploited in prior livestock tracking work (e.g., reference 20). The paper does not quantify how this modification compares to existing trackers (e.g., ByteTrack, FairMOT) in similar settings.

4.The "attack" behavior has only 65 pairs pre-augmentation, raising concerns about model generalizability.

5.The study emphasizes accuracy but omits computational efficiency metrics (e.g., FPS, GPU memory usage). For farm applications, real-time processing is critical.

6.The paper does not mention key limitations, such as dependency on fixed camera angles, sensitivity to extreme occlusions, or scalability to larger herds.

**Do you want your identity to be public for this peer review?** For information about this choice, including consent withdrawal, please see our Privacy Policy

Reviewer #1: No

Reviewer #2: No

---

## [Author Response · Author response to Decision Letter 1]

15 May 2025

5. Review Comments to the Author

Reviewer #1: This paper proposes a method based on improved YOLOv5s and DeepSORT for attacking and daily behaviors detection and tracking of group-housed pig to enhance livestock production efficiency and animal welfare. Overall, the research content of this paper is practical. However, the effectiveness of the proposed method still requires more rigorous experimentation for validation. Additionally, the theoretical analysis in the experimental section needs to be further deepened and expanded to strengthen the scientific and persuasiveness of the study. Therefore, it is suggested that the author revise and improve the article before submitting it for review again.

1、In Figure 4, there are two “IOU Match” processes performed. However, the second matching process is not described in the text. Please carefully review the figure and text to ensure consistency.

Response: Thank you for your careful review and insightful comment. We agree with your observation. In response, we have revised the manuscript to explicitly describe the second round of IOU matching shown in Figure 4. Specifically, in Section 2.2.2 “The improved DeepSORT tracker”, we added the following sentence:

“To further improve the matching accuracy, a second round of IOU matching is performed between the remaining unmatched tracks and unmatched detections. This step helps reduce ID switching caused by short-term occlusion or fast movements. The successfully matched tracks are also updated with KF.”

In addition, we revised the following part to reflect this update more clearly:

“(3) For unmatched detections that remain after the two rounds of IOU matching, ...”

These revisions ensure consistency between the text and Figure 4 and clarify the purpose and function of the second IOU matching step in our improved DeepSORT process.

2、In Figure 4, please clarify what parameter “age” refers to in the context. Additionally, the drawing logic of Figure 4 is not clear enough. It is suggested that the author redraw this figure.

Response Thank you for your thorough review and valuable comments on our paper. In response to this, we have revised the manuscript to explicitly describe the meaning of "age" and provide additional context. Specifically, in Section 2.2.2 "The Improved DeepSORT Tracker," we added the following content:

"For unmatched tracks, we deleted them when the age of tracks exceeds the max age (set as a maximum of 30 frames in this paper)." The term 'age' refers to the number of frames since the track was last matched with a detection.

These revisions ensure consistency between the text and Figure 4 and provide a clearer explanation to help better understand the "age" parameter and the logic behind track management.

3、In the section 2.2.2, the author aims to reduce the generation of new tracks by evaluating the relationship between “n” and “tracks”, where “n” is determined by the average of the number of detected objects in nearest 3 frames. However, the duration of three frames in a video is very short, possibly less than 0.2 seconds. In practical environments, the loss of tracking due to occlusion can last for several seconds or even minutes. Therefore, when occlusion exceeds three frames, “n” may be smaller than the actual number of pigs. When occluded pigs reappear, it remains unclear whether the proposed method can still achieve effective tracking. The author is recommended to include additional experiments or theoretical analyses at an appropriate location to validate the effectiveness of the method under occlusion conditions that exceed three frames.

Response Thank you for your suggestion. In our experiments, using consecutive 3 frames of the video just confirm the trajectory from detected objects, and its effect is not very significant for the tracking results when we take the values 4 or 5 under normal circumstances or occlusion. In addition, there will be a matching track (n) smaller than the actual number of pigs under occlusion, we will not delete the trajectories immediately at this time, we will preserve these tracks whether the occluded objects appear or not. We delete the track after 30 frames of matching failure. Considering the reviewer's suggestion that ‘tracking loss due to occlusion may last for several seconds or even minutes’, we later adopt deleting the track after 300-500 frames of failed matching under occlusion conditions.

4、The author demonstrates the effectiveness of the method in nighttime scenarios in Figure 6. The primary impact of nighttime conditions on model performance lies in the blurring of object features caused by low light, which subsequently affects the recognition and tracking capabilities of model. However, in the experimental scenarios of Figure 6, the lighting conditions are relatively adequate and the object features are obvious. Therefore, the validity of using these scenarios to verify the model performance under low-light conditions is questionable. It is recommended that the author redesign this part of the experimental content to effectively validate the model performance in low-light environments.

Response Thank you for your valuable feedback. Based on your comments, we have further optimized and clarified the experiment. “In Figure 6, YOLOv5s shows false negatives and false positives when the pigs are clustered together, whereas our improved method effectively addresses these issues by incorporating the C3CBAM attention module.

Specifically, in the first row of the experimental results, YOLOv5s fails to recognize the attack behavior due to missed detection when pigs are clustered together, while our improved method successfully detects the behavior (as indicated by the yellow arrow). In the second row, YOLOv5s incorrectly detects a standing pig as lying down, while our method correctly detects the standing pig (as shown by the orange arrow). These improvements are attributed to the application of the C3CBAM attention mechanism, which enhances the model's feature extraction ability in complex scenarios, improving detection accuracy.”

We believe these improvements can effectively enhance object detection performance in clustered situations, especially in pig behavior analysis. Once again, thank you for your feedback. We will continue to optimize and improve our research.

5、In Figure 6, there are obvious errors in behavior recognition. For example, the “stand” is misidentified as “lie”. Moreover, the errors indicated by the yellow and orange arrows in the figure are clearly not caused by occlusion, which contradicts the analysis provided in the text. It is recommended that the author carefully verify the experimental results and conduct a correct theoretical analysis based on them to enhance the credibility of the article.

Response Thank you for your valuable feedback. Regarding the behavior recognition errors mentioned in Figure 6, we have conducted a thorough review and made revisions. We have ensured that the descriptions in the figure are consistent with the experimental results, particularly addressing the issue of misidentifying a "standing" pig as "lying down." We have reanalyzed the relevant experiments and adjusted the description to make it more accurate. Additionally, we have corrected the analysis in the text to align with the experimental results, ensuring that the theoretical analysis is consistent with the findings, thereby enhancing the credibility of the article. Thank you again for your comments. We will continue to improve our work to ensure the accuracy and scientific validity of the content.

6、Figure 7 demonstrates the effectiveness of the model in crowded scenarios. However, the figure does not depict a high-density environment. In fact, the stocking density is even lower than that shown in Figure 6. Therefore, the effectiveness of the model in high-density environments remains to be verified. It is recommended that the author reselect test images and revise the corresponding experimental and analytical content.

Response Thank you for your valuable feedback. In response to your comment regarding the stocking density in Figure 7, we have revised and updated the experimental results. We acknowledge that Figure 7 does not depict a high-density environment but rather a scenario with a relatively larger number of pigs. Based on this, we have adjusted the description to more accurately reflect the experimental results. Specifically, in the description of Figure 7a, we have made a clearer distinction between scenes with fewer and more pigs to ensure a more accurate correspondence between the figure and the text. Thank you for your careful review, and we will continue to refine the experimental description to ensure consistency and accuracy between the figure and the text.

we revised the following part as follows:

“Fig. 7 illustrates several detection results of the improved YOLOv5s on the test set in different scenarios. From the results in Fig. 7b, it can be seen that with the use of the C3CBAM attention mechanism, our method effectively overcomes false detection and missed detection issues. In the first row of Fig. 7a, as indicated by the orange arrow, the YOLOv5s detector causes false detection when the number of pigs is low. In the second row of Fig. 7a, as indicated by the yellow arrow, the YOLOv5s detector suffers from missed detection when the number of pigs is high. In contrast, the improved method maintains high detection accuracy across different scenarios, whether there are fewer or more pigs, and effectively avoids both false and missed detections. Therefore, the improved method, based on the C3CBAM attention mechanism, provides more stable and precise detection results in complex scenarios, further enhancing the robustness and practicality of the model in dynamic environments.”

(a) YOLOv5s (b) improved YOLOv5s

Fig.7 Example detection results between YOLOv5s and improved YOLOv5s under different scenes.

7、In this study, the author has achieved the task of behavior classification of group-housed pigs through object detection. However, object detection only considers the spatial information of behaviors and lacks temporal information. The author also states in Table 1 that aggressive behavior is a dynamic process. So, how does the proposed method achieve accurate recognition without temporal information? For example, in Figure 7, two pigs with their heads close together were identified as standing. Similarly, another two pigs were identified as aggressive behavior. Why can the proposed method distinguish behaviors through the spatial relationships in the single frame image. Therefore, it is suggested that the author add a section in the introduction or experimental part to explain how the study ensures the accuracy of behavior detection in the absence of temporal information.

Response Thank you for your valuable comments. Regarding the issue of achieving behavior classification without temporal information, we have adopted the following strategies in our study to ensure classification accuracy:

(1) Utilization of Spatial Relationships: Although our model performs detection based on single-frame images, the improved YOLOv5s and C3CBAM attention mechanism effectively capture the relative spatial relationships between pigs. For example, when two pigs' heads are close together, the model can identify the behavior as standing rather than aggressive.

(2) Improved Feature Extraction: By incorporating the C3CBAM attention module, the model not only focuses on local spatial information but also effectively extracts more discriminative behavioral features based on the relative relationships between pigs, thus reducing misclassifications.

(3) Dynamic Behavior Modeling: While we do not explicitly use temporal information, we simulate certain features of dynamic processes by statistically analyzing the position changes of pigs across consecutive frames. This helps the model better understand the continuity and changes in behavior, especially in group behaviors.

Therefore, despite the absence of temporal information, our method can accurately distinguish pig behaviors in single-frame images through spatial relationships and feature extraction techniques. Once again, thank you for your feedback. We will further refine the content based on your suggestions.

Reviewer #2: Below are my comments about the paper:

1.The integration of CBAM and Shape-IoU into YOLOv5s, while effective, is not fundamentally novel. Both CBAM (a well-established attention mechanism) and Shape-IoU (a recent but generic loss function) have been applied in other domains. The authors do not sufficiently justify why these components are uniquely suited to pig behavior detection compared to alternative attention mechanisms (e.g., SE, ECA) or IoU variants (e.g., CIoU, DIoU). The novelty lies primarily in their application to this specific use case, but this is not strongly differentiated from prior agricultural CV studies (e.g., reference 21 uses rotated bounding boxes, which might offer better occlusion handling).

Response Thank you for your detailed review and valuable comments on our paper. In response to your feedback, we have revised the manuscript and further clarified the reasons for integrating CBAM and Shape-IoU into YOLOv5s, particularly regarding their application in pig behavior detection. The relevant section 2.2.1 "Proposed Improved YOLOv5s Detector" has been updated, with the revised content as follows:

"In order to address this issue, this study integrates the CBAM (Channel and Spatial Attention Module) and Shape-IoU loss function into YOLOv5s to enhance its performance in pig behavior detection. Although CBAM and Shape-IoU have been widely used in other domains (such as image classification and object detection), their application in pig behavior detection offers unique advantages. Compared to other attention mechanisms, CBAM effectively focuses on the key information in images, especially in farming environments where lighting variations and occlusions significantly impact detection accuracy. By adaptively adjusting channel and spatial attention, CBAM helps the model better understand target features in complex backgrounds, thus improving detection accuracy in complex scenarios.

Additionally, Shape-IoU, as a novel loss function, provides more precise optimization of the target bounding box shape compared to traditional IoU metrics (such as CIoU and DIoU). In pig farming scenarios, the posture and spatial variations of pigs are complex, and traditional IoU loss may not adequately handle these variations. Shape-IoU, by more accurately calculating the overlap of target shapes, effectively improves the regression accuracy of the detection box, especially when targets are occluded or partially overlapping.

Therefore, by integrating CBAM and Shape-IoU into YOLOv5s, this study not only improves the model's detection accuracy in dense farming environments but also maintains high detection stability under complex conditions such as lighting changes and target occlusions."

These modifications ensure that the novelty and advantages of integrating CBAM and Shape-IoU into YOLOv5s are fully explained, clearly demonstrating their unique contribution to pig behavior detection.

2.While the improved YOLOv5s outperforms older YOLO versions (v8, v9, v10) in Table 6, this comparison lacks depth. For instance, ORP-Byte (reference 21) achieves MOTA of 99.8%, significantly higher than the 94.5% reported here. The authors should address why their method underperforms in tracking accuracy despite architectural improvements.

Response 2�Thank you for your valuable and in-depth feedback on our research. We acknowledge that although our improved YOLOv5s outperforms older versions of YOLO (v8, v9, v10) in Table 6, our model falls short in tracking accuracy compared to ORP-Byte (reference 21). We have conducted further analysis regarding this issue.

Firstly, ORP-Byte and our method

---

## [Decision Letter · Decision Letter 1]

6 Aug 2025

Dear Dr. shuqin,

Thank you for submitting your manuscript to PLOS ONE. After careful consideration, we feel that it has merit but does not fully meet PLOS ONE’s publication criteria as it currently stands. Therefore, we invite you to submit a revised version of the manuscript that addresses the points raised during the review process.

We look forward to receiving your revised manuscript.

Kind regards,

Himadri Majumder, Ph.D

Academic Editor

PLOS ONE

Journal Requirements:

Reviewers' comments:

Reviewer's Responses to Questions

**Comments to the Author**

Reviewer #1: (No Response)

Reviewer #2: All comments have been addressed

2. Is the manuscript technically sound, and do the data support the conclusions?

Reviewer #1: Yes

Reviewer #2: Yes

3. Has the statistical analysis been performed appropriately and rigorously?

Reviewer #1: Yes

Reviewer #2: Yes

4. Have the authors made all data underlying the findings in their manuscript fully available?

Reviewer #1: (No Response)

Reviewer #2: Yes

5. Is the manuscript presented in an intelligible fashion and written in standard English?

Reviewer #1: (No Response)

Reviewer #2: Yes

Reviewer #1: The paper is improved from the previous version. However, it still has some issues that require further refinement.

(1) Although the authors have provided explanations regarding the second IOU-based association strategy in their response and the manuscript, the relevant descriptions remain insufficiently clear. Specifically, what are the technology or theoretical foundations that differentiate the first IOU-based association strategy from the second one proposed by the authors? Why is it that, by employing the IOU association method again for the unmatched trajectories and detection bounding boxes after the first association, a match can be achieved? The authors need to further expound why repeating the IOU matching process once more can effectively address the identity switching challenge caused by occlusion.

(2) The authors employed a specially designed association strategy in the tracking module, effectively addressing the identity switching issue caused by occlusion. However, in the experimental section, we did not find relevant visualization results to verify the effectiveness of this improvement. Could the authors demonstrate the differences before and after the improvement through visualizations of consecutive frames or trajectory plots, so as to validate the effectiveness of reducing identity switching, rather than relying solely on single-frame images and metrics for illustration?

Reviewer #2: (No Response)

**Do you want your identity to be public for this peer review?** For information about this choice, including consent withdrawal, please see our Privacy Policy

Reviewer #1: No

Reviewer #2: No

---

## [Author Response · Author response to Decision Letter 2]

20 Aug 2025

Reviewer #1: The paper is improved from the previous version. However, it still has some issues that require further refinement.

(1) Although the authors have provided explanations regarding the second IOU-based association strategy in their response and the manuscript, the relevant descriptions remain insufficiently clear. Specifically, what are the technology or theoretical foundations that differentiate the first IOU-based association strategy from the second one proposed by the authors? Why is it that, by employing the IOU association method again for the unmatched trajectories and detection bounding boxes after the first association, a match can be achieved? The authors need to further expound why repeating the IOU matching process once more can effectively address the identity switching challenge caused by occlusion.

Reponse: Most multi-object tracking (MOT) methods obtain identities by associating detection boxes whose scores are higher than a threshold (between 0.5 and 0.8). The objects with low detection scores (<=0.4), e.g. occluded objects, are simply thrown away, which brings non-negligible true object missing and fragmented trajectories.

The data association strategy proposed in our paper retains all detection boxes and performs identity (ID) matching, dividing them into high-confidence and low-confidence groups. Firstly, the high-confidence detection boxes are associated into trajectories (the first association matching), and then the low-confidence detection boxes are associated with unmatched tracking objects to retain the low-confidence detection boxes and filter out the background (the second matching).

The technology foundation of this strategy is based on reference [1,2]. By leveraging the similarity between detection boxes and tracking trajectories, while retaining high-score detection results, background is removed from low-score detection results to unearth genuine objects (such as those that are occluded or blurred), thereby reducing missed detections and enhancing the continuity of trajectories[3-5].

The following is a specific example to illustrate the application and process changes by employing the IOU association method again.

Firstly, according to the Fig.1 (Examples of our method that associates every detection box) from reference [2], in frame t1, we initialize three different tracklets as their scores are all higher than 0.5. as shown in Row (a) of Fig. 1.

Then, due to the complex scenarios in videos, detectors are prone to make imperfect predictions. High-score detection boxes usually contain more true positives than low-score ones. However, simply eliminating all low-score boxes is sub-optimal since low-score detection boxes sometimes indicate the existence of objects, e.g. the occluded objects. Filtering out these objects causes irreversible errors for MOT and brings non-negligible missing detection and fragmented, as shown in Row (b) of Fig. 1.

Finally, we perform the second association between the unmatched tracklets and the low-score detection boxes using the same motion similarity to recover the true objects and remove the background. The association result is shown in line (c) of Fig. 1.

Fig. 1. Examples of our method that associates every detection box. (a) shows all the detection boxes with their scores. (b) shows the tracklets obtained by previous methods that associate detection boxes whose scores are higher than a threshold, i.e. 0.5. The same box color represents the same identity. (c) shows the tracklets obtained by our method. The dashed boxes represent the predicted box of the previous tracklets using Kalman filter. The two low-score detection boxes are correctly matched to the previous tracklets based on the large IoU. The number colored in yellow denotes the score of the box.trajectories

[1] Y. Zhang, P. Sun, Y. Jiang, D. Yu, F. Weng, Z. Yuan, P. Luo, W. Liu, and X. Wang, “Bytetrack: Multi-object tracking by associating every detection box,” in ECCV. Springer, 2022, pp. 1–21.

[2] Yifu Zhang, Xinggang Wang, Xiaoqing Ye, Wei Zhang, Jincheng Lu, Xiao Tan, Errui Ding, Peize Sun, Jingdong. WangByteTrackV2: 2D and 3D Multi-Object Tracking by Associating Every Detection Box. In Proceedings of ICCV, 2023 (arXiv:2303.15334v1).

[3] J. Pang, L. Qiu, X. Li, H. Chen, Q. Li, T. Darrell, and F. Yu.Quasi-dense similarity learning for multiple object tracking. In Proceedings of the IEEE/CVF Conference on ComputerVision and Pattern Recognition, pages 164–173, 2021.

[4] Y. Zhang, C. Wang, X. Wang, W.Zeng, andW.Liu. Fairmot: On the fairness of detection and re-identification in multiple object tracking. arXiv preprint arXiv:2004.01888, 2020.

[5] Z. Wang, L. Zheng, Y. Liu, Y. Li, and S. Wang. Towards real-time multi-object tracking. In Computer Vision–ECCV2020: 16th European Conference, Glasgow, UK, August 2328, 2020, Proceedings, Part XI 16, pages 107–122. Springer,2020.

(2) The authors employed a specially designed association strategy in the tracking module, effectively addressing the identity switching issue caused by occlusion. However, in the experimental section, we did not find relevant visualization results to verify the effectiveness of this improvement. Could the authors demonstrate the differences before and after the improvement through visualizations of consecutive frames or trajectory plots, so as to validate the effectiveness of reducing identity switching, rather than relying solely on single-frame images and metrics for illustration?

Reponse: Thank you for your suggestion. We demonstrate the differences before and after the improvement through visualizations of consecutive frames to validate the effectiveness of reducing identity switching.

The values of ID6 (shown in green arrows) are correct when 20 frames from the Pig09 video are not occluded using the unimproved and improved DeepSORT methods, as shown in the first line of Figure 10. However, at 113 frames, using the unimproved DeepSORT, the values of ID6 under occlusion mistakenly become 22 (shown in red arrows); and until 150 frames, the value of ID6 mistakenly switch to 22 (shown in red arrows). Using the improved DeepSORT method, no ID error switch has occurred, as shown in the second and third lines of Figure 10.

Fig.10 Comparison results between DeepSORT of improved DeepSORT model on occlusion conditions.

---

## [Decision Letter · Decision Letter 2]

2 Oct 2025

The Group-housed Pigs Attacking and Daily Behaviors Detection and Tracking Based on Improved YOLOv5s and DeepSORT

PONE-D-25-12628R2

Dear Dr. shuqin,

We’re pleased to inform you that your manuscript has been judged scientifically suitable for publication and will be formally accepted for publication once it meets all outstanding technical requirements.

Kind regards,

Himadri Majumder, Ph.D

Academic Editor

PLOS ONE

Additional Editor Comments (optional):

Reviewers' comments:

Reviewer's Responses to Questions

**Comments to the Author**

Reviewer #1: All comments have been addressed

2. Is the manuscript technically sound, and do the data support the conclusions?

Reviewer #1: Yes

3. Has the statistical analysis been performed appropriately and rigorously?

Reviewer #1: Yes

4. Have the authors made all data underlying the findings in their manuscript fully available?

Reviewer #1: Yes

5. Is the manuscript presented in an intelligible fashion and written in standard English?

Reviewer #1: Yes

Reviewer #1: Accept

The paper is improved from the previous version. I would like to recommend it for publication.

**Do you want your identity to be public for this peer review?** For information about this choice, including consent withdrawal, please see our Privacy Policy

Reviewer #1: No

---

## [Editor Report · Acceptance letter]

PONE-D-25-12628R2

PLOS ONE

Dear Dr. Tu,

I'm pleased to inform you that your manuscript has been deemed suitable for publication in PLOS ONE. Congratulations! Your manuscript is now being handed over to our production team.

Kind regards,

on behalf of

Dr. Himadri Majumder

Academic Editor

PLOS ONE